# Collapsing Bandits and Their Application to Public Health Interventions

**Aditya Mate**\*
Harvard University
Cambridge, MA, 02138
aditya_mate@g.harvard.edu

**Jackson A. Killian**\*
Harvard University
Cambridge, MA, 02138
jkillian@g.harvard.edu

**Haifeng Xu**
University of Virginia
Charlottesville, VA, 22903
hx4ad@virginia.edu

**Andrew Perrault**
Harvard University
Cambridge, MA, 02138
aperrault@g.harvard.edu

**Milind Tambe**
Harvard University
Cambridge, MA, 02138
milind_tambe@harvard.edu

## Abstract

We propose and study Collapsing Bandits, a new restless multi-armed bandit (RMAB) setting in which each arm follows a binary-state Markovian process with a special structure: when an arm is played, the state is fully observed, thus "collapsing" any uncertainty, but when an arm is passive, no observation is made, thus allowing uncertainty to evolve. The goal is to keep as many arms in the "good" state as possible by planning a limited budget of actions per round. Such Collapsing Bandits are natural models for many healthcare domains in which health workers must simultaneously monitor patients *and* deliver interventions in a way that maximizes the health of their patient cohort. Our main contributions are as follows: (i) Building on the Whittle index technique for RMABs, we derive conditions under which the Collapsing Bandits problem is *indexable*. Our derivation hinges on novel conditions that characterize when the optimal policies may take the form of either "forward" or "reverse" threshold policies. (ii) We exploit the optimality of threshold policies to build fast algorithms for computing the Whittle index, including a closed form. (iii) We evaluate our algorithm on several data distributions including data from a real-world healthcare task in which a worker must monitor and deliver interventions to maximize their patients' adherence to tuberculosis medication. Our algorithm achieves a 3-order-of-magnitude speedup compared to state-of-the-art RMAB techniques, while achieving similar performance. The code is available at: https://github.com/AdityaMate/collapsing_bandits

## 1 Introduction

**Motivation.** This paper considers scheduling problems in which a planner must act on $k$ out of $N$ binary-state processes each round. The planner fully observes the state of the processes on which she acts, then all processes undergo an action-dependent Markovian state transition; the state of the process is unobserved until it is acted upon again, resulting in uncertainty. The planner's goal is to maximize the number of processes that are in some "good" state over the course of $T$ rounds. This class of problems is natural in the context of *monitoring tasks* which arise in many domains such as sensor/machine maintenance [12, 10, 1, 31], anti-poaching patrols [26], and especially healthcare. For example, nurses or community health workers are employed to monitor and improve the adherence of patient cohorts to medications for diseases like diabetes [23], hypertension [4], tuberculosis [27, 5]

---

and HIV [16, 15]. Their goal is to keep patients adherent (i.e., in the "good" state) but a health worker can only intervene on (visit) a limited number of patients each day. Health workers can play a similar role in monitoring and delivering interventions for patient mental health, e.g., in the context of depression [20, 22] or Alzheimer's Disease [18].

We adopt the solution framework of *Restless Multi-Arm Bandits* (RMABs), a generalization of Multi-Arm Bandits (MABs) in which a planner may act on $k$ out of $N$ arms each round that each follow a Markov Decision Process (MDP). Solving an RMAB is PSPACE-hard in general [24]. Therefore, a common approach is to consider the Lagrangian relaxation of the problem in which the $\frac{k}{N}$ budget constraint is dualized. Solving the relaxed problem gives Lagrange multipliers which act as a greedy index heuristic, known as the Whittle index, for the original problem. Specifically, the *Whittle index policy* computes the Whittle index for each arm, then plays the top $k$ arms with the largest indices. The Whittle index policy has been shown to be asymptotically optimal (i.e., $N \to \infty$ with fixed $\frac{k}{N}$) under a technical condition [32] and generally performs well empirically [3] making it a common solution technique for RMABs.

Critically, using the Whittle index policy requires two key components: (i) a fast method for computing the index and (ii) proving the problem satisfies a technical condition known as *indexability*. Without (i) the approach can be prohibitively slow, and without (ii) asymptotic performance guarantees are sacrificed [32]. Neither (i) nor (ii) are known for general RMABs. Therefore, to capture the scheduling problems addressed in this work, we introduce a new subclass of RMABs, *Collapsing Bandits*, distinguished by the following feature: when an arm is played, the agent fully observes its state, "collapsing" any uncertainty, but when an arm is passive, no observation is made and uncertainty evolves. We show that this RMAB subclass is more general than previous models and leads to new theoretical results, including conditions under which the problem is indexable and under which optimal policies follow one of two simple threshold types. We use these results to develop algorithms for quickly computing the Whittle index. In experiments, we analyze the algorithms' performance on (i) data from a real-world healthcare scheduling task in which our approach ties state-of-the-art performance at a fraction the runtime and (ii) various synthetic distributions, some of which the algorithm achieves performance comparable to the state of the art even outside its optimality conditions.

To summarize, our contributions are as follows: (i) We introduce a new subclass of RMABs, Collapsing Bandits, (ii) Derive theoretical conditions for Whittle indexability and for the optimal policy to be threshold-type, and (iii) Develop an efficient solution that achieves a 3-order-of-magnitude speedup compared to more general state-of-the-art RMAB techniques, without sacrificing performance.

## 2 Restless Multi-Armed Bandits

An RMAB consists of a set of $N$ arms, each associated with a *two-action* MDP [25]. An MDP $\{\mathcal{S}, \mathcal{A}, r, P\}$ consists of a set of states $\mathcal{S}$, a set of actions $\mathcal{A}$, a state-dependent reward function $r : \mathcal{S} \to \mathbb{R}$, and a transition function $P$, where $P_{s,s'}^a$ denotes the probability of transitioning from state $s$ to $s'$ when action $a$ is taken. An MDP *policy* $\pi : \mathcal{S} \to \mathcal{A}$ represents a choice of action to take at each state. We will consider both discounted and average reward criteria. The long-term *discounted reward* starting from state $s_0 = s$ is defined as $R_\beta^\pi(s) = E\left[\sum_{t=0}^\infty \beta^t r(s_{t+1} \sim T(s_t, \pi(s_t), s_{t+1}) | \pi, s_0 = s\right]$ where $\beta \in [0, 1)$ is the discount factor and actions are selected using $\pi$. To define average reward, let $f^\pi(s) : \mathcal{S} \to [0, 1]$ denote the *occupancy frequency* induced by policy $\pi$, i.e., the fraction of time spent in each state of the MDP. The *average reward* $\overline{R}^\pi$ of policy $\pi$ be defined as the expected reward computed over the occupancy frequency: $\overline{R}^\pi = \sum_{s \in \mathcal{S}} f^\pi(s) r(s)$.

Each arm in an RMAB is an MDP with the action set $\mathcal{A} = \{0, 1\}$. Action 1 (0) is called the *active* (*passive*) action and denotes the arm being pulled (not pulled). The agent can pull at most $k$ arms at each time step. The agent's goal is to maximize either her discounted or average reward across the arms over time. Some RMAB problems need to account for partial observability of states. It is sufficient to let the MDP state be the *belief state*: the probability of being in each latent state [14]. While intractable in general due to infinite number of reachable belief states, most partially observable RMABs studied (including our Collapsing Bandits) have polynomially many belief states due to a finite time horizon or other structures.

**Related work** RMABs have been an attractive framework for studying various stochastic scheduling problems since Whittle indices were introduced [34]. Because general RMABs are PSPACE-hard [24], RMAB studies usually consider restricted classes under which some performance guarantees can be derived. Collapsing Bandits form one such novel class that generalizes some existing results which we note in later sections. Liu and Zhao [19] develop an efficient Whittle index policy for a 2-state partially observable RMAB subclass in which the state transitions are unaffected by the actions taken and reward is accrued from the active arms only. Akbarzadeh and Mahajan [2] define a class of bandits with "controlled restarts," giving indexability results and a method for computing the Whittle index. However, "controlled restarts" define the active action as state independent, a stronger assumption than Collapsing Bandits which allow state-dependent action effects. Glazebrook et al. [10] give Whittle indexability results for three classes of restless bandits: (1) A machine maintenance regime with deterministic active action effect (we consider stochastic active action effect) (2) A switching regime in which the passive action freezes state transitions (in our setting, states always change regardless of action) (3) A reward depletion/replenishment bandit which deterministically resets to a start state on passive action (we consider stochastic passive action effect). Hsu [11] and Sombabu et al. [29] augment the machine maintenance problem from Glazebrook et al. [10] to include either i.i.d. or Markovian evolving probabilities of an active action having no effect, a limited form of state-dependent action. Meshram et al. [21] introduce Hidden Markov Bandits which, similar to our approach, consider binary state transitions under partial observability, but do not allow for state dependent rewards on passive arms. In sum, our Collapsing Bandits introduce a new, more general RMAB formulation than special subclasses previously considered. Qian et al. [26] present a generic approach for any indexable RMAB based on solving the (partially observable) MDPs on arms directly. Because we derive a closed form for the Whittle index, our algorithm is orders of magnitude faster.

## 3 Collapsing Bandits

We introduce *Collapsing Bandits* (CoB) as a specially structured RMAB with partial observability. In CoB, each arm $n \in \{1, \ldots, N\}$ has binary latent states $\mathcal{S} = \{0, 1\}$, representing *bad* and *good* state, respectively. The agent acts during each of finite days $t \in 1, \ldots, T$. Let $a_t \in \{0, 1\}^N$ denote the vector of actions taken by the agent on day $t$. Arm $n$ is said to be *active* at $t$ if $a_t(n) = 1$ and *passive* otherwise. The agent acts on $k$ arms per day, i.e., $\|a_t\| = k$, where $k \ll N$ because resources are limited. When acting on arm $n$, the true latent state of $n$ is fully observed by the agent and thus its uncertainty "collapses" to a realization of the binary latent state. We denote this observation as $\omega \in \mathcal{S}$. States of passive arms are completely unobservable by the agent.

Active arms transition according to the *transition matrix* $P_{s,s'}^{a,n}$ and passive arms transition according to $P_{s,s'}^{p,n}$. We drop the superscript $n$ when there is no ambiguity. Our scheduling problem, like many problems in analogous domains, exhibits the following natural structure: (i) processes are more likely to stay "good" than change from "bad" to "good"; (ii) when acted on, they tend to improve. These natural structures are respectively captured by imposing the following constraints on $P^p$ and $P^a$ for each arm: (i) $P_{0,1}^p < P_{1,1}^p$ and $P_{0,1}^a < P_{1,1}^a$; (ii) $P_{0,1}^p < P_{0,1}^a$ and $P_{1,1}^p < P_{1,1}^a$. To avoid unnecessary complication through edge cases, all transition probabilities are assumed to be nonzero. The agent receives reward

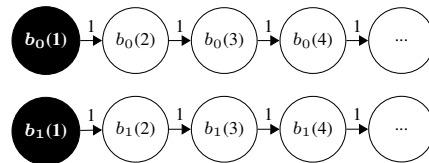

Figure 1: Belief-state MDP under the policy of always being passive. There is one chain for each observation $\omega \in \{0, 1\}$ with the head marked black. Belief states deterministically transition down the chains.

$r_t = \sum_{n=1}^N s_t(n)$ at $t$, where $s_t(n)$ is the latent state of arm $n$ at $t$. The agent's goal is to maximize the long term rewards, either discounted or average, defined in Sec. 2.

**Belief-State MDP Representation** In limited observability settings, belief-state MDPs have organized chain-like structures, which we will exploit. In particular, the only information that affects our belief of an arm being in state 1 is the number of days since that arm was last pulled and the state $\omega$ observed at that time. Therefore, we can arrange these belief states into two "chains" of length $T$, each for an observation $\omega$. A sketch of the belief state chains under the passive action is shown in

Fig. 1. Let $b_\omega(u)$ denote the belief state, *i.e., the probability that the state is* $1$, if the agent received observation $\omega \in \{0, 1\}$ when it acted on the process $u$ days ago. Note that $b_\omega(u)$ is also the expected reward associated with that belief state, and let $\mathcal{B}$ be the set of all belief states.

When the belief-state MDP is allowed to evolve under some policy, the following mechanism arises: first, after an action, the state $\omega$ is observed (uncertainty "collapses"), then one round passes causing the agent's belief to become $P_{\omega,1}^a$, representing the head of the chain determined by $\omega$. Subsequent passive actions cause the process to transition deterministically down the same chain (though, the transition in the latent state is still stochastic). Then when the process's arm is active, it transitions to the head of one of the chains with probability equal to the belief that the corresponding observation would be emitted (see Fig. 2a for an illustration).

The belief associated with a belief state can be calculated in closed form with the given transition probabilities. Formally,

$$b_\omega(u) = \tau_{u-1}(P_{\omega,1}^a)\ \forall u \in [T] \text{ where } \tau_u(b) = \frac{P_{0,1}^p - (P_{1,1}^p - P_{0,1}^p)^u(P_{0,1}^p - b(1 + P_{0,1}^p - P_{1,1}^p))}{(1 + P_{0,1}^p - P_{1,1}^p)} \quad (1)$$

## 4 Collapsing Bandits: Threshold Policies and Whittle Indexability

Because of the well-known intractability of solving general RMABs, the widely adopted solution concept in the literature of RMABs is the Whittle index approach; for a comprehensive description, see Whittle [34]. Intuitively, the Whittle index captures the value of acting on an arm in a particular state by finding the minimum *subsidy* $m$ the agent would accept to *not act*, where the subsidy is some exogenous "donation" of reward. Formally, the modified reward function becomes $r_m : \mathcal{S} \times \mathcal{A} \to \mathbb{R}$, where $r_m(s, 0) = r(s) + m$ and $r_m(s, 1) = r(s)$. Let $R_{\beta,m}^\pi(s) = E\left[\sum_{t=0}^\infty \beta^t r_m(s_t, \pi(s_t))|\pi, s_0 = s\right]$ and $\overline{R}_m^\pi = \sum_{s \in \mathcal{S}} f^\pi(s) r_m(s, \pi(s))$ be the discounted and average reward criteria for this new subsidy setting, respectively. The former is maximized by the discounted value function (we give a value function for the average reward criterion in **Fast Whittle Index Computation**):

$$V_m(b) = \max \begin{cases} m + b + \beta V_m(\tau_1(b)) & \text{passive} \\ b + \beta(bV_m(P_{1,1}^a) + (1-b)V_m(P_{0,1}^a)) & \text{active} \end{cases} \quad (2)$$

where $\tau$ is defined in Eq. 1 and $b$ is shorthand for $b_\omega(u)$. In a CoB, the Whittle index of a belief state $b$ is the smallest $m$ s.t. it is equally optimal to be active or passive in the current state. Formally:

$$W(b) = \inf_m \{m : V_m(b; a = 0) \geq V_m(b; a = 1)\} \quad (3)$$

Critically, performance guarantees hold only if the problem satisfies *indexability* [32, 34], a condition which says that for all states, the optimal action cannot switch to active as $m$ increases. Let $\Pi_m^*$ be the set of policies that maximize a given reward criterion under subsidy $m$.

**Definition 1** (Indexability). *An arm is indexable if $\mathcal{B}^*(m) = \{b : \forall \pi \in \Pi_m^*, \pi(b) = 0\}$ monotonically increases from $\emptyset$ to the entire state space as $m$ increases from $-\infty$ to $\infty$. An RMAB is indexable if every arm is indexable.*

The following special type of MDP policy is central to our analysis.

**Definition 2** (Threshold Policies). *A policy is a* forward (reverse) threshold policy *if there exists a threshold $b_{th}$ such that $\pi(b) = 0$ ($\pi(b) = 1$) if $b > b_{th}$ and $\pi(b) = 1$ ($\pi(b) = 0$) otherwise.*

**Theorem 1.** *If for each arm and any subsidy $m \in \mathbb{R}$, there exists an optimal policy that is a forward or reverse threshold policy, the Collapsing Bandit is indexable under discounted and average reward criteria.*

*Proof Sketch.* Using linearity of the value function in subsidy $m$ for any fixed policy, we first argue that when forward (reverse) threshold policies are optimal, proving indexability reduces to showing that the threshold monotonically decreases (increases) with $m$. Unfortunately, establishing such a monotonic relationship between the threshold and $m$ is a well-known challenging task in the literature that often involves problem-specific reasoning [19]. Our proof features a sophisticated induction argument exploiting the finite size of $\mathcal{B}$ and relies on tools from real analysis for limit arguments.

$\square$

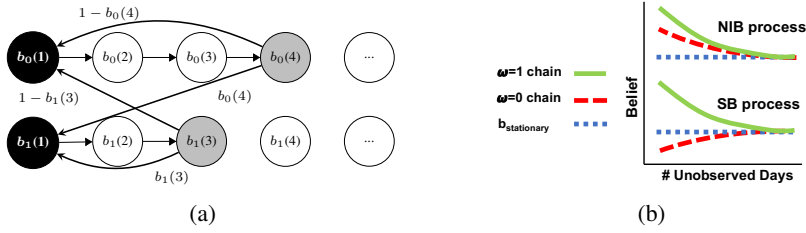

(a)                                                (b)

Figure 2: (a) Visualization of forward threshold policy ($X_0 = 4, X_1 = 3$). Black nodes are the head of each chain and grey nodes are the thresholds. (b) Non-increasing belief (NIB) process has non-increasing belief in both chains. A split belief process (SB) has non-increasing belief after being observed in state 1, but non-decreasing belief after being observed in state 0.

All formal proofs can be found in the appendix. We remark that Thm. 1 generalizes the result in the seminal work by Liu and Zhao [19] who proved the indexability for a special class of CoB. In particular, the RMAB in Liu and Zhao [19] can be viewed as a CoB setting with $P^a = P^p$, i.e., transitions are independent of actions.

Though the Whittle index is known to be challenging to compute in general [34], we are able to design an algorithm that computes the Whittle index efficiently assuming the optimality of threshold policies, which we now describe.

**Fast Whittle Index Computation**  The main algorithmic idea we use is the Markov chain structure that arises from imposing a *forward* threshold policy on an MDP. A forward threshold policy can be defined by a tuple of the first belief state in each chain that is less than or equal to some belief threshold $b_{th} \in [0, 1]$. In the two-observation setting we consider, this is a tuple $(X_0^{b_{th}}, X_1^{b_{th}})$, where $X_\omega^{b_{th}} \in 1, \ldots, T$ is the index of the first belief state in each chain where it is optimal to act (i.e., the belief is less than or equal to $b_{th}$). We now drop the superscript $b_{th}$ for ease of exposition. See Fig. 2a for a visualization of the transitions induced by such an example policy. For a forward threshold policy $(X_0, X_1)$, the occupancy frequencies induced for each state $b_\omega(u)$ are:

$$f^{(X_0,X_1)}(b_\omega(u)) = \begin{cases} \alpha & \text{if } \omega = 0, u \leq X_0 \\ \beta & \text{if } \omega = 1, u \leq X_1 \\ 0 & \text{otherwise} \end{cases} \tag{4}$$

$$\alpha = \left( \frac{(X_1 b_0(X_0))}{1 - b_1(X_1)} + X_0 \right)^{-1}, \beta = \left( \frac{X_1 b_0(X_0)}{1 - b_1(X_1)} + X_0 \right)^{-1} \frac{b_0(X_0)}{1 - b_1(X_1)} \tag{5}$$

These equations are derived from standard Markov chain theory. These occupancy frequencies do not depend on the subsidy. Let $J_m^{(X_0,X_1)}$ be the average reward of policy $(X_0, X_1)$ under subsidy $m$. We decompose the average reward into the contribution of the state reward and the subsidy

$$J_m^{(X_0,X_1)} = \sum_{b \in \mathcal{B}} b f^{(X_0,X_1)}(b) + m(1 - f^{(X_0,X_1)}(b_1(X_1)) - f^{(X_0,X_1)}(b_0(X_0))) \tag{6}$$

Recall that for any belief state $b_\omega(u)$, the Whittle index is the smallest $m$ for which the active and passive actions are both optimal. Given forward threshold optimality, this translates to two corresponding threshold policies being equally optimal. Such policies must have adjacent belief states as thresholds, as can be concluded from Lemma 1 in Appendix 7. Note that for a belief state $b_0(X_0)$ the only adjacent threshold policies with active and passive as optimal actions at $b_0(X_0)$ are $(X_0, X_1)$ and $(X_0 + 1, X_1)$ respectively. Thus the subsidy which makes these two policies equal in value must thus be the Whittle Index for $b_0(X_0)$, which we obtain by solving: $J_m^{(X_0,X_1)} = J_m^{(X_0+1,X_1)}$ for $m$. We use this idea to construct two fast Whittle index algorithms.

**Sequential index computation algorithm**  Alg. 1 precomputes the Whittle index of every belief state for each process, having time complexity $\mathcal{O}(|\mathcal{S}|^2 T N)$. Then, the per-round complexity to retrieve the top $k$ indices is $\mathcal{O}(N \min\{k, log(N)\})$. This gives a great improvement over the more general method given by Qian et al. [26] (our main competitor) which has per-round complexity

of $\approx \mathcal{O}(N \log(\frac{1}{\epsilon})(|\mathcal{S}|T)^{2+\frac{1}{18}})$, where $\log(\frac{1}{\epsilon})$ is due to a bifurcation method for approximating the Whittle index to within error $\epsilon$ on each arm and $(|\mathcal{S}|T)^{2+\frac{1}{18}}$ is due to the best-known complexity of solving a linear program with $|\mathcal{S}|T$ variables [13].

Alg. 1 is optimized for settings in which the Whittle index can be precomputed. However, for online learning settings, we give an alternative method in Appendix 12 that computes the Whittle index on-demand, in a closed form.

---

**Algorithm 1:** Sequential index computation algorithm

---

Initialize counters to heads of the chains: $X_1 = 1$, $X_0 = 1$
**while** $X_1 < T$ *or* $X_0 < T$ **do**
    Compute $m_1 := m$ such that $J_m^{(X_0,X_1)} = J_m^{(X_0,X_1+1)}$
    Compute $m_0 := m$ such that $J_m^{(X_0,X_1)} = J_m^{(X_0+1,X_1)}$
    Set $i = \arg\min\{m_0, m_1\}$ and $W(X_i) = \min\{m_0, m_1\}$
    Increment $X_i$
**end**

---

Our algorithm also requires that belief is decreasing in $X_0$ and $X_1$. Formally, we require:

**Definition 3** (Non-increasing belief (NIB) processes)**.** *A process has* non-increasing belief *if, for any* $u \in [T]$ *and for any* $\omega \in \mathcal{S}$, $b_\omega(u) \geq b_\omega(u+1)$.

All possible CoB belief trends are shown in Fig. 2b. We make this distinction because the computation of the Whittle index in Alg. 1 is guaranteed to be exact for NIB processes that are also forward threshold optimal, though we show empirically that our approach works surprisingly well for most distributions. In the next section, we analyze the possible forms of optimal policies to find conditions under which threshold policies are optimal.

**Types of Optimal Policies** Analyzing Eq. 2 reveals that at most three types of optimal policies exist. This follows directly from the definition of $V_m(b)$, which is a max over the passive action value function and the active action value function. The former is convex in $b$, a well-known POMDP result [30], and the latter is linear in $b$. Thus, as shown in Fig. 3, there are three ways in which the value functions of each action may intersect; this defines three optimal policy forms of *forward*, *reverse* and *dual* threshold types, respectively. Forward and reverse threshold policies are defined in Def. 2; dual threshold policies are active between two

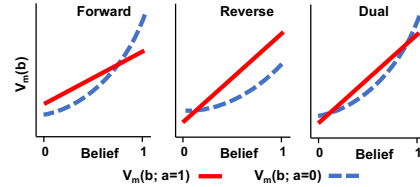

Figure 3: Components of $V_m(b)$ in Eq. 2. Since the passive action is convex in $b$, active action is linear in $b$, and value function is a max over these, at most three optimal policy types are possible.

separate threshold points and passive elsewhere. Not only do threshold policies greatly reduce the optimization search space, they often admit closed form expressions for the index as demonstrated earlier in this section. We now derive sufficient conditions on the state transition probabilities under which each type of policy is verifiably optimal.

**Theorem 2.** *Consider a belief-state MDP corresponding to an arm in a Collapsing Bandit. For any subsidy $m$, there is a* forward *threshold policy that is optimal under the condition:*

$$(P_{1,1}^p - P_{0,1}^p)(1 + \beta(P_{1,1}^a - P_{0,1}^a))(1 - \beta) \geq P_{1,1}^a - P_{0,1}^a \tag{7}$$

*Proof Sketch.* Forward threshold optimality requires that if the optimal action at a belief $b$ is passive, then it must be so for all $b' > b$. This can be established by requiring that the derivative of the passive action value function is greater than the derivative of the active action value function w.r.t. $b$. The main challenge is to distill this requirement down to measurable quantities so the final condition can be easily verified. We accomplish this by leveraging properties of $\tau(b)$ and using induction to derive both upper and lower bounds on $V_m(b_1) - V_m(b_2) \; \forall \, b_1, b_2$ as well as a lower bound on $\frac{d(V_m(b))}{db}$. $\qquad\square$

Intuitively, the condition requires that the intervention effect on processes in the "bad" state must be large, making $P_{1,1}^a - P_{0,1}^a$ small. Note that Liu and Zhao [19] consider the case where $P_{1,1}^a = P_{1,1}^p$ and $P_{0,1}^a = P_{0,1}^p$, which makes Eq. 7 always true. Thus we generalize their result for threshold optimality.

**Theorem 3.** *Consider a belief-state MDP corresponding to an arm in a Collapsing Bandit. For any subsidy m, there is a* reverse *threshold policy that is optimal under the condition:*

$$(P_{1,1}^p - P_{0,1}^p)\Big(1 + \frac{\beta(P_{1,1}^a - P_{0,1}^a)}{1 - \beta}\Big) \le P_{1,1}^a - P_{0,1}^a \tag{8}$$

Intuitively, the condition requires small intervention effect on processes in the "bad" state, the opposite of the forward threshold optimal requirement. Note that both Thm. 2 and Thm. 3 also serve as conditions for the average reward case as $\beta \to 1$ (a proof based on Dutta's Theorem [8] is given in Appendix 10).

**Conjecture 1.** *Dual threshold policies are never optimal for Collapsing Bandits.*

This conjecture is supported by extensive numerical simulations over the random space of state transition probabilities, values of $\beta$, and values of subsidy $m$; its proof remains an open problem. Note that this would imply that all Collapsing Bandits are indexable.

## 5 Experimental Evaluation

We evaluate our algorithm on several domains using both real and synthetic data distributions. We test the following algorithms: **Threshold Whittle** is the algorithm developed in this paper. **Qian et al. [26]**, a slow, but precise general method for computing the Whittle index, is our main baseline that we improve upon. **Random** selects $k$ process to act on at random each round. **Myopic** acts on the $k$ processes that maximize the expected reward at the immediate next time step. Formally, at time $t$, this policy picks the $k$ processes with the largest values of $\Delta b_t = (b_{t+1}|a = 1) - (b_{t+1}|a = 0)$. **Oracle** fully observes all states and uses Qian et al. [26] to calculate Whittle indices. We measure performance in terms of *intervention benefit*, where $0\%$ corresponds to the reward of a policy that is always passive and $100\%$ corresponds to Oracle. All results are averaged over 50 independent trials.

### 5.1 Real Data: Monitoring Tuberculosis Medication Adherence

We first test on tuberculosis medication adherence monitoring data, which contains daily adherence information recorded for each real patient in the system, as obtained from Killian et al. [17]. The "good" and "bad" states of the arm (patient) correspond to "Adhering" and "Not Adhering" to medication, respectively. State transition probabilities are estimated from the data. Because this data is noisy and contains only the adherence records and not the intervention (action) information (as the authors state), we perturb the computed average transition matrix by reducing (increasing) $P_{\omega,1}$ by a uniform random number between 0 and $\delta_1, \delta_2$ ($\delta_3, \delta_4$) then renormalizing to obtain $P_{\omega,1}^p$ ($P_{\omega,1}^a$) for the simulation. Reward is measured as the undiscounted sum of patients (arms) in the adherent state over all rounds, where each trial lasts $T = 180$ days (matching the length of first-line TB treatment) with $N$ patients and a budget of $k$ calls per day. All experiments in this section set all $\delta$ to 0.05.

In Fig. 4a, we plot the runtime in seconds vs the number of patients $N$. Fig. 4b compares the intervention benefit for $N = 100, 200, 300, 500$ patients and $k = 10\%$ of $N$. In the $N = 200$ case, the runtimes of a single trial of Qian et al. and Threshold Whittle index policy are 3708 seconds and 3 seconds, respectively, while attaining near-identical intervention benefit. Our algorithm is thus 3 orders of magnitude faster than the previous state of the art without sacrificing performance.

We next test Threshold Whittle as the resource level $k$ is varied. Fig. 4c shows the performance in the $k = 5\%N$, $k = 10\%N$ and $k = 15\%N$ regimes ($N = 200$). Threshold Whittle outperforms Myopic and Random by a large margin in these low resource settings. We also affirm the robustness of our algorithm to $\delta$, the perturbation parameter used to approximate real-world $P_{\omega,1}^p$ and $P_{\omega,1}^a$ from the data, and present the extensive sensitivity analysis in Appendix 13. Finally, in Appendix 12 we couple our algorithm to a Thompson Sampling-based learning approach and show it performs well in the real-world case where transition probabilities would need to be learned online, supporting the deployability of our work.

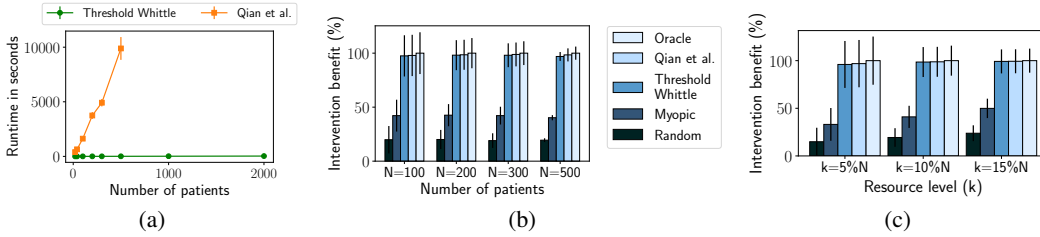

Figure 4: (a) Threshold Whittle is several orders of magnitude faster than Qian et al. and scales to thousands of patients without sacrificing performance on realistic data (b). (c) Intervention benefit of Threshold Whittle is far larger than naive baselines and nearly as large as Oracle.

## 5.2 Synthetic Domains

We test our algorithm on four synthetic domains, that potentially characterize other healthcare or relevant domains, and highlight different phenomena. Specifically, we: (i) identify situations when Myopic fails completely while Whittle remains close to optimal, (ii) analyze the effect of latent state entropy on policy performance, (iii) identify limitations of Threshold Whittle by constructing processes for which Threshold Whittle shows separation from Oracle, and (iv) test robustness of our algorithm outside of the theoretically guaranteed conditions. To facilitate comparison with the real data distribution, we simulate trials for $T = 180$ rounds where reward is the undiscounted sum of arms in state 1 over all rounds. We consider the space of transition probabilities satisfying the assumed natural constraints, as outlined in Sec. 3.

Fig. 5a demonstrates a domain characterized by processes that are either self-correcting or non-recoverable. Self-correcting processes have a high probability of transitioning from state 0 to 1 regardless of the action taken, while non-recoverable processes have a low chance of doing so. We show that when the immediate reward is larger for the former than the latter, Myopic can perform even worse than Random. That is because a myopic policy always prefers to act on the self-correcting processes per their larger immediate reward, while Threshold Whittle, capable of long-term planning, looks to avoid spending resources on these processes. In this regime, the best long-term plan is to always act on the non-recoverable processes to keep them from failing. Analytical explanation of this phenomenon is presented in Appendix 11. We set the resource level, $k = 10\%N$ in our simulation for Fig. 5a. Note that performance of Myopic drops as the fraction of self-correcting processes becomes larger and reaches a minimum at $x = 100\% - k = 90\%$. Beyond this point, Threshold Whittle can no longer completely avoid self-correcting processes and the gap subsequently starts to decrease.

Fig. 5b explores the effect of uncertainty in the latent state on long-term planning. For each point on the $x$-axis, we draw all transition probabilities according to $P_{\omega,1}^p, P_{\omega,1}^a \sim [x, x+0.1]$. The entropy of the state of a process is maximum near 0.5 making long term planning most uncertain and as a result, this point shows the biggest gap with Oracle, which can observe all the states in each round. Note that Myopic and Whittle policies perform similarly, as expected for (nearly) stochastically identical arms.

Fig. 5c studies processes that have a large propensity to transition to state 0 when passive and a corresponding low active action impact, but a significantly larger active action impact in state 1. This makes it attractive to exclusively act on processes in the 1 state. This simulates healthcare domains where a fraction of patients degrade rapidly, but can recover, and indeed respond very well to interventions if already in a good state. To simulate these, we draw transition matrices with $P_{0,1}^p, P_{1,1}^p, P_{0,1}^a \sim [0.3, 0.32]$ and $P_{1,1}^a \sim [0.7, 0.72]$ in varying proportions and sample the rest from the real TB adherence data. Because the best plan is to act on processes in state 1, both Myopic and Whittle act on the processes with the largest belief giving Oracle a significant advantage as it has perfect knowledge of states.

Although we provide theoretical guarantees on our algorithm for forward threshold optimal processes with non-increasing belief, Fig. 5d reveals that Alg. 1 performs well empirically even with these conditions relaxed. Here, we sample processes uniformly at random from the state transition probability space, and use rejection sampling to vary the proportion of threshold optimal processes. Threshold Whittle performs well even when as few as $20\%$ of the processes are forward threshold optimal; we briefly analyze this phenomenon in Appendix 14.

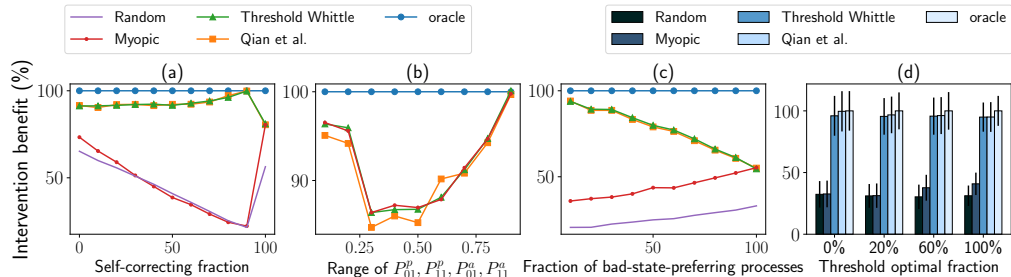

Figure 5: (a) Myopic can be trapped into performing even worse than Random while Threshold Whittle remains close to optimal. (b) Long-term planning is least effective when entropy of states is maximum. (c) Myopic and Whittle planning become similar when more processes are prone to failures. (d) Threshold Whittle is surprisingly robust to processes even outside of theoretically guaranteed conditions.

## 6  Conclusion

We open a new subspace of Restless Bandits, *Collapsing Bandits*, which applies to a broad range of real-world problems, especially in healthcare delivery. We give new theoretical results that cover a large portion of real-world data as well as an algorithm that runs thousands of times faster than the state of the art without sacrificing performance. We simultaneously also recognize limitations of our theoretical results, which become narrow in the average reward case. We envision several interesting avenues for future work, including techniques to incorporate the user/health worker inputs for planning, generalizing our inherently 2-state approach to allow for a multi-state model, and allowing multiple actions and/or more general reward functions.

## Broader Impact

Our work is largely motivated by resource constrained health intervention delivery. This setting is common across low, middle, and high-income countries, in which community health workers (CHWs) are recruited to deliver basic care to a cohort of patients or benefactors. In fact, CHWs have been critical in achieving global health initiatives for over five decades, and evidence shows that CHWs have had a positive impact in myriad domains including maternal and newborn health [6, 9], (non-)communicable diseases [6, 28], and sexual/reproductive health [35]

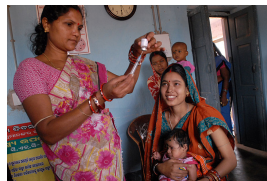

Figure 6: CHW delivering vaccine. Credit: Pippa Ranger.

in low-resource communities across the world [7, 9, 28, 33]. Our modeling has the potential to improve the delivery of care in these highly resource-constrained settings.

However, a deployment of our system to any setting must be done responsibly. For instance, we designed our system with the intention of *assisting* human CHWs plan resource-limited interventions. That said, we present results that highlight our algorithm's ability to plan for thousands of processes at a time, far more than for which a human could independently plan. Just making this capability available could encourage the automation of applicable interventions via automated calls or texts, potentially displacing CHW jobs, reducing human contact with patients, and unfairly limiting care for patients with limited access to technology.

Additionally, users of the system must be dutifully aware that its recommendations will be based solely on the data entered in the system. In the context of medication adherence monitoring, if the worker enters incorrect data, e.g., the patient was adhering ("good" state) but they instead mark the patient as not adhering ("bad" state), then the algorithm could make the wrong recommendation about the patient the next day, since its belief of the patient's adherence would also be wrong.

Finally, our AI system is inherently a blackbox which would likely be replacing an interpretable scheduling heuristic. This would limit any user or administrator's ability to audit decisions around

why certain patients were recommended for intervention. As with any potential deployment of a blackbox system to a domain that affects the allocation of resources to humans, system designers should be acutely aware of the balance between their needs to be able to perform audits vs. their need for optimization.

## Acknowledgments and Disclosure of Funding

This work was supported in part by the Army Research Office by grant Multidisciplinary University Research Initiative (MURI) grant number W911NF1810208. J.A.K. was supported by an NSF Graduate Research Fellowship under Grant DGE1745303. A.P. was supported by the Harvard Center for Research on Computation and Society.

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
