[Supplementary Material]

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

## Footnotes

[2] For simplicity, this assumes the starting belief is equal to the belief at the head of one of the chains, i.e., $P_{1,1}^a$ or $P_{0,1}^a$. However, we could add to the set $\mathcal{B}$ another $T$ belief states corresponding to a chain that starts from any arbitrary belief and evolves for $T$ passive actions. These new states could be ordered appropriately within $\mathcal{B}$ and the rest of the proof would follow unchanged.

[3] For reverse threshold optimal processes, simply arrange $\mathcal{B}$ and $\Pi$ in ascending order of belief. The rest of the proof follows similarly.

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

# Appendix

## 7    Proof of Indexability

We give the proof assuming forward threshold policies are optimal, and note where relevant how the proof also works for reverse threshold optimal policies.

**Fact 1.** *For two non-concurrent, increasing, linear functions $f_1(m)$ and $f_2(m)$ and two points $m_1, m_2$, such that $m_1 \leq m_2$, if $f_1(m_1) \leq f_2(m_1)$ and $f_1(m_2) \geq f_2(m_2)$, then $\frac{df_1}{dm} \geq \frac{df_2}{dm}$. Additionally, if $f_1(m_1) < f_2(m_1)$ and $f_1(m_2) \geq f_2(m_2)$, then $\frac{df_1}{dm} > \frac{df_2}{dm}$.*

*Proof.* We now start proving the theorem by assuming that forward belief threshold policies are optimal. Let $b_{th}^*(m)$ denote the threshold corresponding to the optimal threshold policy for a given $m$. To show indexability, we must show that if a belief state $b$ is passive, i.e., $b > b_{th}^*(m_1)$, for some $m_1$, then it is also passive, i.e., $b > b_{th}^*(m_2)$, for all $m_2 \geq m_1$.

In our problem, we have $2T$ belief states which, for a forward threshold policy, can be arranged in a descending order of their belief values: $\mathcal{B} := \{b_{2T}, b_{2T-1}, \ldots, b_i, \ldots, b_1\}$.[2] A forward threshold policy is then any real value $b_{th}$ which splits $\mathcal{B}$ into a passive set $\mathcal{P} = \{b_i : b_i > b_{th} \; \forall b_i \in \mathcal{B}\}$ and active set $\mathcal{C} = \{b_i : b_{th} \geq b_i \; \forall b_i \in \mathcal{B}\}$. Note that all values of $b_{th}$ such that $b_{i+1} \geq b_{th} > b_i$ $\forall i \in 1, \ldots, 2T$ correspond to the same threshold policy. Thus there are only $2T + 1$ unique threshold policies possible corresponding to the $2T + 1$ such belief regions marked by points in $\mathcal{B}$. Let $\Pi = \{\pi_{2T+1}, \pi_{2T}, \ldots, \pi_1\}$ denote these unique possible threshold policies arranged in a decreasing order, where $\pi_i \geq \pi_j$ implies $b_{th}^*(\pi_i) \geq b_{th}^*(\pi_j)$ where $b_{th}^*(\pi_i)$ is the optimal belief threshold associated with $\pi_i$.[3] Thus the threshold policy $\pi_i$ would follow: $b_i > b_{th}^*(\pi_i) \geq b_{i-1}$ $\forall i \in 1, \ldots, 2T + 1$, where $b_0 := -\infty$ and $b_{2T+1} := \infty$. Note that in a policy $\pi_i$, if for a belief state $b$, the optimal action is passive, then under a policy $\pi_j$, the optimal action at $b$ is also passive $\forall j \leq i$ because $b_{th}^*(\pi_i) \geq b_{th}^*(\pi_j)$. Thus to prove indexability, it is sufficient to show that:

$$\begin{aligned}
&\forall m_1, m_2 \text{ such that } m_1 \leq m_2, \\
&\text{if } \pi^*(m_1) = \pi_i \text{ and } \pi^*(m_2) = \pi_j, \text{ then} \\
&\implies i \geq j
\end{aligned} \tag{9}$$

where $\pi^*(m)$ denotes the optimal threshold policy at subsidy $m$.

**Lemma 1.** Let $m_i^*$ be the *infimum* among all $m$'s for which $\pi^*(m) = \pi_i$. Then, the infimum is achievable (i.e., $\pi^*(m_i^*) = \pi_i$) and moreover $m_{2T+1}^* < m_{2T}^* < \ldots < m_1^*$.

*Proof.* We prove this using induction. Consider the base case: $m_{2T+1}^* < m_i^*$ $\forall i < 2T + 1$. When $m \to -\infty$, the optimal action would clearly always be to act to avoid accruing large negative reward. So $\pi_{2T+1}$ would be the optimal policy for $m \to -\infty$ and clearly the base case is true.

For the inductive case, assume the hypothesis, $m_{2T+1}^* < \ldots < m_{t+1}^* < m_i^* \; \forall i < t + 1$. Let $m_t^*$ be the *infimum* among all $m$'s for which $\pi^*(m) = \pi_t$. We must show: (1) $m_t^* < m_i^* \; \forall i < t$; (2) $\pi^*(m_t^*) = \pi_t$ (i.e., the infimum is achievable). For convenience, we denote $L = \{\pi_t, \pi_{t-1}, \ldots \pi_1\}$ as the set of "lower-side" polices and $U = \{\pi_{2T+1}, \pi_{2T}, \ldots \pi_{t+1}\}$ as the set of "upper-side" policies.

As $m$ is increased beyond $m_{t+1}^*$, let $m'$ be the *infimum value* among all $m$'s whose optimal policy is from $L = \{\pi_t, \pi_{t-1}, \ldots \pi_1\}$ (note, the definition of $m'$ is different from $m_t^*$ since at this point we do not know whether the smallest $m$'s optimal policy is $\pi_t$ or some $\pi_i$ with $i < t$ yet). That is, either the optimal threshold policy at $m'$ is from $L$ (when the infimum is achievable) or there exists an infinite sequence $\{\bar{m}_l\}_{l=1}^\infty$ that converges *from the right side* to $m'$ (i.e., $\bar{m}_l \geq m'$ for all $s$) and the optimal policy for any $\bar{m}_l$ is from policy set $L$ (when the infimum is not achievable). For notational convenience, we will think of the former achievable case also as that there is a sequence $\{\bar{m}_l\}_{l=1}^\infty$

that converges to $m'$ and the optimal policy for any $\bar{m}_l$ is from $L$ (letting all $\bar{m}_l = m'$ will do). In fact, a stronger conclusion holds. That is, we can choose an infinite-length sequence $\{\bar{m}_l\}_{l=1}^{\infty}$ such that the optimal policy for each $\bar{m}_l$ will be the same. This simply follows from the fact that $\{\bar{m}_l\}_{l=1}^{\infty}$ has infinite length, and their optimal policy is from a finite set $L$. So some policy from $L$ must be optimal for infinitely many of $\bar{m}_l$'s. Therefore, we shall assume that $\bar{m}_l \to m'$ from the right side and the optimal policy for each $\bar{m}_l$ is some $\bar{\pi} \in L$.

Our main claim is that for subsidy $m'$, the passive action and active action must both be optimal at state $b_t$. Therefore, by definition, this implies the threshold policy $\pi_t$ is optimal for $m'$. We thus have $m_t^* = m'$, $m_i^* > m_t^*$ $\forall i < t$, and moreover $\pi_t$ is indeed optimal for $m_t^*$ (i.e., the infimum is achievable). This concludes the induction proof. The remainder of this proof will be devoted to prove this claim.

By definition of $m'$, there exists a sequence $\{\underline{m}_u\}_{u=1}^{\infty}$ that converges to $m'$ *from the left side* (i.e., $\underline{m}_u < m'$ for all $t$) and moreover the optimal policy for any $\underline{m}_u$ is from the policy set $U = \{\pi_{2T+1}, \pi_{2T}, ...\pi_{t+1}\}$. Similar to the above reasoning, we shall choose the sequence $\{\underline{m}_u\}_{u=1}^{\infty}$ such that their optimal policy is the same $\underline{\pi} \in U$.

We now prove that the passive action and active action must both be optimal at state $b_t$ for $m'$. Assume, for the sake of contradiction, that the optimal action at $b_t$ for subsidy $m'$ is passive and that the active action is not optimal (the other case where the optimal action is active follows a similar contradiction argument). That means the optimal policy for $m'$ has a threshold $b_{th}^*(m') < b_t$ and thus $\pi^*(m') \in L$. Moreover, since the active action is not optimal for $b_t$, $\underline{\pi}$ must not be optimal for $m'$ and thus achieves strictly less reward than $\pi^*(m')$. Since $\underline{m}_u \to m'$, we thus have

$$\lim_{u \to \infty} V_{\underline{m}_u}(\underline{\pi}) = V_{m'}(\underline{\pi}) < V_{m'}(\pi(m')),$$

where the last inequality uses the fact that $\underline{\pi}$ is sub-optimal for $m'$ because the active action is strictly sub-optimal for $b_t$. On the other hand,

$$V_{m'}(\pi(m')) = \lim_{u \to \infty} V_{\underline{m}_u}(\pi(m')) \leq \lim_{u \to \infty} V_{\underline{m}_u}(\underline{\pi})$$

These two inequalities contradict each other. This concludes our proof of the lemma. $\qquad\square$

Let $\pi_i$ be the optimal policy at some $m_1$.

$$\implies m_i^* \leq m_1$$
$$\implies m_j^* < m_i^* \leq m_1 \ \forall j > i \text{ using Lemma 1}$$

Let $V_\pi(m, b)$ be the discounted reward of policy $\pi$ at arbitrary state $b$ as defined in Eq. 2 of the main text. Then for any $V_{\pi_i}(m, b)$ and $V_{\pi_j}(m, b)$ such that $j > i$ we have:

$$V_{\pi_i}(m_j^*, b) < V_{\pi_j}(m_j^*, b) \ (\pi_j \text{ is optimal at } m_j^*) \tag{10}$$
$$V_{\pi_i}(m_i^*, b) \geq V_{\pi_j}(m_i^*, b) \ (\pi_i \text{ is optimal at } m_i^*) \tag{11}$$
$$m_j^* < m_i^* \text{ if } j > i \tag{12}$$
$$\implies \frac{dV_{\pi_i}}{dm} > \frac{dV_{\pi_j}}{dm} \forall j > i \tag{13}$$

Where Eq. 10 is a strict inequality as implied by Lemma 1 and Eq. 13 follows from Fact 1 and the value function's linear dependence on $m$ (whether discounted or average reward criterion). We now claim that $\forall m_j > m_i^*$, if $\pi_j$ is optimal for $m_j$ then we must have $j \leq i$. Towards a contradiction, assume $j > i$. Then similar to the above equations, we have the following:

$$V_{\pi_i}(m_j, b) \leq V_{\pi_j}(m_j, b) \ (\pi_j \text{ is optimal at } m_j) \tag{14}$$
$$V_{\pi_i}(m_i^*, b) \geq V_{\pi_j}(m_i^*, b) \ (\pi_i \text{ is optimal at } m_i^*) \tag{15}$$
$$m_i^* < m_j \tag{16}$$
$$\implies \frac{dV_{\pi_i}}{dm} \leq \frac{dV_{\pi_j}}{dm} \forall j > i \tag{17}$$

Where Eq. 17 follows from Fact 1 and the value function's linear dependence on $m$ (whether discounted or average reward criterion). which contradicts Eq. 13. Therefore, our claim holds. From 9, that implies indexability. $\qquad\square$

# 8 Technical Condition for Forward Threshold Policies to be Optimal

We restate Eq. 2 here:

$$V_m(b) = max \begin{cases} m + b + \beta V_m(\tau(b)) & \text{passive} \\ b + \beta(bV_m(P_{1,1}^a) + (1-b)V_m(P_{0,1}^a)) & \text{active} \end{cases}$$

where $\tau(b) := \tau_1(b)$ from Eq. 1. Simplified, $\tau(b)$ is simply a linear function of $b$ given by the expression

$$\begin{aligned} \tau(b) &= bP_{1,1}^p + (1-b)P_{0,1}^p \\ &= (P_{1,1}^p - P_{0,1}^p)b + P_{0,1}^p \end{aligned} \tag{18}$$

We will start by stating two facts, then proving three useful technical lemmas.

**Fact 2.** $\frac{d(\tau(b))}{db} = (P_{1,1}^p - P_{0,1}^p) \leq 1$.

**Fact 3.** $\forall b, b'$ s.t. $b \geq b'$, $\tau(b) \geq \tau(b')$.

Facts 2 and 3 follow from Eq 18.

**Lemma 2.** $V_m(b_1) - V_m(b_2) \geq b_1 - b_2, \forall b_1, b_2$ s.t. $b_1 > b_2$

*Proof.* We will proceed via induction, where the base case will be a one-step value function. Then we will show that the t-step value function assumption implies the t+1-step inductive value function hypothesis. In the base case the value function equals only the one-step immediate reward. It is sufficient to compare the value functions $V_m^1(b_1)$ and $V_m^1(b_2)$ element-wise, since if the true optimal action for one of the value functions is passive and the other active, the bound can still be established by flipping the action of one of the value functions as needed. This gives:

Base case $V_m^1(b_1) - V_m^1(b_2) =$

$$m + b_1 - (m + b_2) = b_1 - b_2 \qquad \text{passive} \tag{19}$$
$$b_1 - b_2 = b_1 - b_2 \qquad \text{active} \tag{20}$$

is clearly satisfied. Now assume $V_m^t(b_1) - V_m^t(b_2)) \geq b_1 - b_2$. Then $V_m^{t+1}(b_1) - V_m^{t+1}(b_2)$

Case 1 (both passive):

$$\begin{aligned} &= m + b_1 + \beta V_m^t(\tau(b_1)) - (m + b_2 + \beta V_m^t(\tau(b_2))) \\ &= b_1 - b_2 + \beta\Big(V_m^t(\tau(b_1)) - V_m^t(\tau(b_2))\Big) \\ &\geq b_1 - b_2 + \beta(\tau(b_1) - \tau(b_2)) \\ &\geq b_1 - b_2 \end{aligned} \tag{21}$$

Case 2 (both active):

$$\begin{aligned} &= b_1 - b_2 + \beta\Big((b_1 - b_2)V_m^t(P_{1,1}^a) + (b_2 - b_1)V_m^t(P_{0,1}^a)\Big) \\ &= b_1 - b_2 + \beta\Big((b_1 - b_2)(V_m^t(P_{1,1}^a) - V_m^t(P_{0,1}^a))\Big) \\ &= (b_1 - b_2)(1 + \beta(V_m^t(P_{1,1}^a) - V_m^t(P_{0,1}^a)) \\ &\geq (b_1 - b_2)(1 + \beta * 0) \\ &= (b_1 - b_2) \end{aligned} \tag{22}$$

$\square$

**Corollary 1.** $V_m(b)$ is an increasing function in $b$, i.e., $V_m(b) \geq V_m(b'), \forall b, b'$ s.t. $b \geq b'$.

*Proof.* The proof follows from Lemma 2 by setting $b_1 = b$ and $b_2 = b'$. $\square$

**Lemma 3.** $V_m(b_1) - V_m(b_2) \leq \frac{b_1 - b_2}{1-\beta}, \forall b_1, b_2$ s.t. $b_1 > b_2$

*Proof.* Proceed by induction again. The base case $V_m(b_1) - V_m(b_2) =$

$$m + b_1 - (m + b_2) = b_1 - b_2 \leq \frac{b_1 - b_2}{1 - \beta} \qquad \text{both passive} \qquad (23)$$

$$b_1 - b_2 = b_1 - b_2 \leq \frac{b_1 - b_2}{1 - \beta} \qquad \text{both active} \qquad (24)$$

which are both clearly satisfied. Now assume $V_m^t(b_1) - V_m^t(b_2) \leq \frac{b_1 - b_2}{1 - \beta}$. Then, $V_m^{t+1}(b_1) - V_m^{t+1}(b_2)$

Case 1 (both passive):

$$
\begin{aligned}
&= \big(m + b_1 + \beta V_m^t(\tau(b_1))\big) - \big(m + b_2 + \beta V_m^t(\tau(b_2))\big) \\
&= (b_1 - b_2) + \beta\big(V_m^t(\tau(b_1)) - V_m^t(\tau(b_2))\big) \\
&\leq (b_1 - b_2) + \beta\left(\frac{\tau(b_1) - \tau(b_2)}{1 - \beta}\right) \\
&\leq (b_1 - b_2) + \beta\left(\frac{(b_1 - b_2)}{1 - \beta}\right) \text{ by Fact 3} \\
&= \frac{b_1 - b_2}{1 - \beta}
\end{aligned}
\qquad (25)
$$

Case 2 (both active):

$$
\begin{aligned}
&= \Big(b_1 + \beta\big(b_1 V_m^t(P_{1,1}^a) + (1 - b_1)V_m^t(P_{0,1}^a)\big)\Big) - \\
&\quad \Big(b_2 + \beta\big(b_2 V_m^t(P_{1,1}^a) + (1 - b_2)V_m^t(P_{0,1}^a)\big)\Big) \\
&= (b_1 - b_2) + \beta\Big((b_1 - b_2)\big(V_m^t(P_{1,1}^a) - V_m^t(P_{0,1}^a)\big)\Big) \\
&\leq (b_1 - b_2) + \beta\left((b_1 - b_2).\frac{P_{1,1}^a - P_{0,1}^a}{1 - \beta}\right) \\
&\leq (b_1 - b_2) + \beta\left(\frac{(b_1 - b_2)}{1 - \beta}\right) \text{ by Fact 2} \\
&= \frac{b_1 - b_2}{1 - \beta}
\end{aligned}
\qquad (26)
$$

$\square$

**Lemma 4.** $\frac{d(V_m(b))}{db} \geq 1 + \beta\alpha$
where, $\alpha = \min\{P_{1,1}^p - P_{0,1}^p, P_{1,1}^a - P_{0,1}^a\}$

*Proof.* Using Eq. 2, we get:

$$
\frac{d(V_m(b))}{db} = \begin{cases} 1 + \beta\frac{d(V_m(\tau(b)))}{d(\tau(b))}\frac{d(\tau(b))}{db} & \text{passive} \\ 1 + \beta(V_m(P_{1,1}^a) - V_m(P_{0,1}^a)) & \text{active} \end{cases}
\qquad (27)
$$

Case 1 (passive):

$$= 1 + \beta\frac{d(V_m(\tau(b)))}{d(\tau(b))}(P_{1,1}^p - P_{0,1}^p) \qquad (28)$$

$$= 1 + \beta \lim_{\delta \to 0} \frac{V_m(\tau(b) + \delta) - V_m(\tau(b))}{\tau(b) + \delta - \tau(b)}(P_{1,1}^p - P_{0,1}^p) \qquad (29)$$

$$\geq 1 + \beta(P_{1,1}^p - P_{0,1}^p) \text{ by Lemma 2} \qquad (30)$$

$$\geq 1 + \beta\alpha \qquad (31)$$

Case 2 (active):

$$= 1 + \beta(V_m(P_{1,1}^a) - V_m(P_{0,1}^a)) \tag{32}$$

$$\geq 1 + \beta(P_{1,1}^a - P_{0,1}^a) \text{ by Lemma 2} \tag{33}$$

$$\geq 1 + \beta\alpha \tag{34}$$

$\square$

Now we derive the technical condition for **Theorem 2**. In this case, proving that threshold policies are optimal is equivalent to proving that, if it is optimal to act now, then it is optimal to act for all later beliefs. Formally, if for a belief $b$, the optimal action is to act, then we must show that for a lower $b' < b$, the optimal action is also to act. To do this, we show that Theorem 2 implies that the derivative wrt $b$ of the passive action value function is greater than the derivative wrt $b$ of the active action value function:

$$(P_{1,1}^p - P_{0,1}^p)(1 + \beta(P_{1,1}^a - P_{0,1}^a))(1 - \beta) \geq P_{1,1}^a - P_{0,1}^a \tag{35}$$

Note that since $(P_{1,1}^a - P_{0,1}^a) \leq 1, \implies (1 + \beta(P_{1,1}^a - P_{0,1}^a))(1 - \beta) \leq 1$, Eq.35 itself implies that $\alpha = P_{1,1}^a - P_{0,1}^a$. Thus, it becomes:

$$(P_{1,1}^p - P_{0,1}^p)(1 + \beta\alpha)(1 - \beta) \geq P_{1,1}^a - P_{0,1}^a \tag{36}$$

$$\implies (P_{1,1}^p - P_{0,1}^p)(1 + \beta\alpha) \geq V_m(P_{1,1}^a) - V_m(P_{0,1}^a) \text{ by Lemma 3} \tag{37}$$

$$\implies (P_{1,1}^p - P_{0,1}^p)\frac{d(V_m(b))}{db} \geq V_m(P_{1,1}^a) - V_m(P_{0,1}^a) \text{ by Lemma. 4} \tag{38}$$

$$\implies 1 + \beta\frac{d(V_m(\tau(b)))}{d(\tau b)}\frac{d(\tau(b))}{db} \geq 1 + \beta(V_m(P_{1,1}^a) - V_m(P_{0,1}^a)) \text{ by Fact 2} \tag{39}$$

$$\implies \frac{d(V_m(b|a=0))}{d(b)} \geq \frac{d(V_m(b|a=1))}{d(b)} \tag{40}$$

$$\tag{41}$$

# 9 Technical Condition for Reverse Threshold Policies to be Optimal

Now we derive a technical condition for a reverse threshold policy. That is, a threshold policy in which if it is optimal to be passive in the current state, then it must also be optimal to act in all later states in the order. First we prove one more technical Lemma.

**Lemma 5.** $\frac{d(V_m(b))}{db} \leq 1 + \frac{\beta\gamma}{1-\beta}$
where, $\gamma = \max\{P_{1,1}^p - P_{0,1}^p, P_{1,1}^a - P_{0,1}^a\}$

*Proof.* Using Equation 8, we get:

$$\frac{d(V_m(b))}{db} = \begin{cases} 1 + \beta\frac{d(V_m(\tau(b)))}{d(\tau(b))}\frac{d(\tau(b))}{db} & \text{passive} \\ 1 + \beta(V_m(P_{1,1}^a) - V_m(P_{0,1}^a)) & \text{active} \end{cases} \tag{42}$$

Case 1 (passive):

$$= 1 + \beta\frac{d(V_m(\tau(b)))}{d(\tau(b))}(P_{1,1}^p - P_{0,1}^p) \tag{43}$$

$$= 1 + \beta\lim_{\delta \to 0}\frac{V_m(\tau(b) + \delta) - V_m(\tau(b))}{\tau(b) + \delta - \tau(b)}(P_{1,1}^p - P_{0,1}^p) \tag{44}$$

$$\leq 1 + \frac{\beta}{1-\beta}(P_{1,1}^p - P_{0,1}^p) \text{ by Lemma 3} \tag{45}$$

$$\leq 1 + \frac{\beta\gamma}{1-\beta} \tag{46}$$

Case 2 (active):

$$= 1 + \beta(V_m(P_{1,1}^a) - V_m(P_{0,1}^a)) \tag{47}$$

$$\leq 1 + \frac{\beta}{1-\beta}(P_{1,1}^a - P_{0,1}^a) \text{ by Lemma 3} \tag{48}$$

$$\leq 1 + \frac{\beta\gamma}{1-\beta} \tag{49}$$

$$\square$$

Now to give a condition under which reverse threshold policies are optimal. Formally, if for a belief $b$, the optimal action is to be passive, then we must show that for a lower $b' < b$, the optimal action is also to be passive. We do this by showing that the Theorem 3 statement implies that the derivative wrt $b$ of the passive value function is less than the derivative wrt $b$ of the active action value function:

$$(P_{1,1}^p - P_{0,1}^p)(1 + \frac{\beta(P_{1,1}^a - P_{0,1}^a)}{1-\beta}) \leq P_{1,1}^a - P_{0,1}^a \tag{50}$$

Note that the Eq. 50 itself implies that $\gamma = P_{1,1}^a - P_{0,1}^a$, thus giving:

$$(P_{1,1}^p - P_{0,1}^p)(1 + \frac{\beta\gamma}{1-\beta}) \leq P_{1,1}^a - P_{0,1}^a \tag{51}$$

$$\implies (P_{1,1}^p - P_{0,1}^p)(1 + \frac{\beta\gamma}{1-\beta}) \leq V_m(P_{1,1}^a) - V_m(P_{0,1}^a) \text{ by Lemma 2} \tag{52}$$

$$\implies (P_{1,1}^p - P_{0,1}^p)\frac{d(V_m(b))}{db} \leq V_m(P_{1,1}^a) - V_m(P_{0,1}^a) \text{ by Lemma 5} \tag{53}$$

$$\implies 1 + \beta\frac{d(V_m(\tau(b)))}{d(\tau b)}\frac{d(\tau(b))}{db} \leq 1 + \beta(V_m(P_{1,1}^a) - V_m(P_{0,1}^a)) \text{ by Fact 2} \tag{54}$$

$$\implies \frac{d(V_m(b|a=0))}{d(b)} \leq \frac{d(V_m(b|a=1))}{d(b)} \tag{55}$$

$$\tag{56}$$

## 10  Threshold Conditions for Average Reward Case

First we define the concept of *value boundedness* [8]:

**Definition 4** (Value Boundedness). *For a given MDP, consider a value function $V_\beta(b)$, states $b \in \mathcal{B}$ and some index state $z \in \mathcal{B}$. Then an MDP is value bounded if for a constant $M$ and function $M(b)$:*

$$M(b) < V_\beta(b) - V_\beta(z) < M \tag{57}$$

We now prove that Thm. 2 and Thm. 3 hold respectively under the average reward criterion as $\beta \to 1$ using Dutta's Theorem as follows [8]. Consider an MDP that is *value bounded*. Let $\pi_\beta(\cdot)$ be a stationary optimal policy for the discounted MDP. (1) Suppose $\pi_\beta(\cdot) \to \pi$ pointwise, as $\beta \to 1$. Then $\pi$ is a stationary optimal policy for the average reward criterion. (2) Furthermore, given state ordering $O$, if for all discounted optimal policies $\pi_\beta(b)$, $O(b') \geq O(b)$ implies $\pi_\beta(b') \geq \pi_\beta(b)$ (i.e., threshold policies are optimal), then any sequence of discounted optimal policies converge to an average optimal policy as $\beta \to 1$.

(2) and (1) together imply that any MDP that admits threshold optimal policies under discounted reward criteria also admits threshold optimal policies under average reward criteria. By construction, any MDP that satisfies Thm. 2 or Thm. 3 admits threshold optimal policies under the discounted reward criterion. Therefore, to prove that those conditions hold under the average reward criterion as $\beta \to 1$, we need only prove that any CoB is value bounded.

**Theorem 4.** *Any Collapsing Bandit is value bounded.*

## 11 Example When the Myopic Policy Fails

We present an example in which the myopic baseline is barely better than No Calls, while Threshold Whittle is *optimal*. Consider the system with $N = 2$ and $k = 1$ and the transition probabilities shown in Fig. 7a.

$$P^{p,1} = \begin{bmatrix} 0.97 & 0.03 \\ 0.03 & 0.97 \end{bmatrix} \quad P^{a,1} = \begin{bmatrix} 0.96 & 0.04 \\ 0.01 & 0.99 \end{bmatrix}$$

$$P^{p,2} = \begin{bmatrix} 0.25 & 0.75 \\ 0.03 & 0.97 \end{bmatrix} \quad P^{a,2} = \begin{bmatrix} 0.23 & 0.77 \\ 0.01 & 0.99 \end{bmatrix}$$

(a)           (b)

Figure 7: For the example transition matrices, Myopic performs worse than random, while Threshold Whittle is nearly optimal.

Fig. 7b shows how various policies perform on these two processes. The myopic policy is worse than random and threshold Whittle is nearly optimal. The myopic policy always acts on process 2 because the immediate reward it considers, $(b_{t+1}|a = 1) - (b_{t+1}|a = 0)$ is marginally higher for process 2 than process 1. However, process 1 is better to pull in the long run because process 2 has a large $P^p_{0,1}$, making it self-correcting, meaning the process is likely to become adhering quickly even without an intervention. However, process 1 has a very small $P^a_{0,1}$ and $P^p_{0,1}$ and is thus difficult to revive from the bad state even with an intervention, making it important to keep intervening to stop the process from ever entering the bad state.

The following analysis shows that the myopic policy always prefers to pull arm 2:

For process 1:

$$
\begin{aligned}
(b_{t+1}|a = 0) &= 0.97.b_t + 0.03.(1 - b_t) & &= 0.94.b_t + 0.03 \\
(b_{t+1}|a = 1) &= 0.99.b_t + 0.01.(1 - b_t) & &= 0.95.b_t + 0.04 \\
\text{Thus, } \Delta b_t = (b_{t+1}|a = 1) - (b_{t+1}|a = 0) & & &= 0.01 + 0.01.b_t < 0.02
\end{aligned}
$$

Similarly, for process 2:

$$\Delta b_t = 0.02$$

The myopic policy chooses the arm with the greater $\Delta b_t$.

## 12 Learning Online

So far we assumed that *all transition probabilities are known*. However, in a real deployment, the transition probabilities of processes would be unknown at the start, and it would be desirable to learn the transition probabilities online in tandem with planning. To develop an online planning regime for our algorithm, we use the tuberculosis medication adherence monitoring domain from the main text as a case study and motivating example.

We implement a Thompson sampling-based learning method [32], which is a heuristic which has been shown to work well in practice and has been frequently used in the bandit literature [14]. In Thompson sampling, we sample from a posterior distribution over the estimated parameters and use the samples for planning. This allows for "sub-optimal" actions to be taken periodically, building exploration implicitly into planning. Then, as arms are pulled we use the observations to update our posterior distribution. We maintain a Beta distribution posterior over the parameters of each row of a patient's transition matrix and sample from it each day to generate a matrix with which the system can plan for that round.

Additionally, we consider two specific features of the TB medication adherence monitoring domain that can be used to accelerate learning with Thompson sampling. First, it is reasonable to assume that

patients (processes) might remember some number of previous days of their medication adherence behavior. Thus, when the agent pulls an arm, the arm may reveal state observations for some number of previous days which we call *buffer length*. The larger the buffer length, the faster learning will converge since more observations are obtained for updating the posterior. We parameterize buffer length and evaluate its effect on learning and planning in experiments. Second, we verify with real data that virtually all patients adhere to the natural constraints on the transition probabilities given in Section 3. We exploit this known structure on the transition probabilities – i.e., that processes tend to degrade when passive and that interventions must have positive effect – to identify a constrained probability space from which we would like to sample when learning online. We implement a version of Thompson sampling called *constrained* Thompson sampling which samples from this joint, constrained probability space via rejection sampling.

**On-demand index computation algorithm.** When we learn online, the transition matrices for a process change every day, and thus pre-computing the Whittle indices for every belief state as in Alg. 1 is inefficient. We can address this by identifying and solving only the indifference equation that is relevant to the current state of the process. We use the insight that for a threshold of $X_i$ on the current chain $i$, the corresponding threshold $X_j$ on chain $j$ would be the state with the largest belief lower than $b(X_i)$, i.e., $X_j = \min_u \{u : b_j(u) < b(X_i)\}$. The Whittle index for $X_i$ is then obtained by solving for $m : J_m^{(X_i, X_j)} = J_m^{(X_i+1, X_j)}$. These computations are repeated every day yielding overall complexity of $\mathcal{O}(|\Omega| T^2)$ per process.

Figure 8: (left) Constrained Thompson sampling improves learning. (right) buffer lengths of 4–7 perform well for various values of $k/N$, using constrained Thompson sampling. TW_X is the on-demand index algorithm run in tandem with Thompson sampling and a buffer length of X.

Fig. 8 (right) evaluates the impact of varying buffer lengths for various ratios of $k/N$. Note that in these experiments, Oracle fully observes states, but must still learn transition probabilities online. Critically, we see that even when simulated patients report 4–7 observations per arm-pull, the performance is close to that of the non-Oracle learning upper bound (buffer length=$\infty$) for any $k/N$. This is a key consideration for deployment in a medication adherence context: patients need only remember their last 4–7 doses on average for our approach to be nearly effective as possible in the TB context.

Fig. 8 (left) compares the performance of learning policies with and without constrained Thompson sampling for $k/N = 25\%$. All policies benefit from the constrained sampling approach, suggesting that imposing our knowledge of the transition probability constraints was beneficial to learning.

## 13  Sensitivity Analysis

In Fig. 9, we investigate Threshold Whittle's performance relative to the choice of parameters used to perturb the real data from the TB medication adherence domain. All the plots show that Threshold Whittle's performance is robust to the choice of parameters.

Figure 9: Performance of Threshold Whittle is robust to perturbation of the transition matrix parameters. Note that 100% corresponds to the performance of Threshold Whittle for this plot only.

Figure 10: (a) Threshold Whittle-computed indices vs. reachable beliefs for 10 randomly sampled reverse threshold optimal processes (one line per process). These indices tend to increase in belief, as expected for reverse threshold optimal processes according to the proof in Appendix 7. (b) Threshold Whittle-computed indices vs. reachable beliefs for 10 randomly sampled forward threshold optimal processes (one line per process). These indices always decrease in belief, as expected for forward threshold optimal processes according to the proof in Appendix 7.

## 14 Threshold Whittle's Performance on Reverse Threshold Optimal Processes

Here we investigate why Threshold Whittle demonstrates near-optimal performance even on reverse-threshold-optimal processes. We randomly sample forward and reverse threshold optimal processes, checked with Thm. 2 and Thm. 3, respectively, using $\beta = 0.95$, then compute their indices with the Threshold Whittle algorithm. Figures. 10a and 10b show a few samples of these trajectories for reverse and forward threshold optimal processes, respectively. Via similar arguments from the proof in Appendix 7, it can be shown that the true Whittle indices for reverse (forward) threshold

optimal processes should always be increasing (decreasing) in belief. Fig. 10a shows that for such reverse threshold optimal processes, the indices computed by Threshold Whittle do tend to increase in belief as expected, which may lead to Threshold Whittle's good performance even though it is not guaranteed to be optimal on those processes. (And for completeness, Fig. 10b shows that for forward threshold optimal policies, the indices computed by Threshold Whittle always decrease in belief as expected.)