[Reviews · NeurIPS 2020]

Review 1

Summary and Contributions: The paper studies a new version of the restless bandits problem, where each are corresponds to an MDP and their state changes if the agent decides to take an action for them. At every time step the agent can take actions on a limited number of arms due limited resources. Furthermore, the agent receives reward from all the arm according to their state. In their version of this restless bandit problem, the authors assume that agent gets to observe the state of the arms that she acts on, hence, collapsing any uncertainty for the active arms. The authors' motivation comes from healthcare applications. They provide conditions under which the MDP of each arm is indexable and when its optimal policy takes the form of a forward or reverse policy. They further provide a fast algorithm for computing the Whittle index as well as a closed for it. Finally they verify their findings on a semi-synthetic data.

Strengths: - (significance: high) The authors study a new version of RMAB that has not been addressed in the literature before. It is definitely very valuable to the researchers. - (significance: high) They provide very neat results regarding identifying conditions under which the proposed problem is indexible and when it can tackled using simple "forward" and "reverse" policies. The conditions seem to be fairly general and simple. - (significance: medium) The authors propose a fast algorithm for computing index in the proposed setting. They also derive a close form for index.

Weaknesses: - The policy is not described fully clearly. It seems that the Whittle Threshold policy can be either performed using forward or reverse policies. How should we decide which one to incorporate? It is also not clear which one is being used in the simulations. Also, how do we choose which k arms out N arms to act on given their indexes? - The simulation part has some ambiguity. If I understand correctly your proposed index computation policy is exact similar to Qina et al. But why is it the case that their policy outperforms yours?

Correctness: The proof sketches seems to be correct and the results are consistent and make sense. However I did not check the appendix to see the full proofs.

Clarity: The paper is vey well-written without any typos (at least I could not find any). The proof sketches could be a little more intuitive though. Also, I would rather see a clean section describing the policy and clean separate section explaining the policy along with the theoretical analysis. Right now, they are entangled together and it is hard to follow.

Relation to Prior Work: The authors have a done a good job interns of explaining how their model and results differs from the previous works.

Reproducibility: Yes

Additional Feedback: Regarding the motivation of the problem, the authors suggest healthcare applications. Why is it the case that in these applications one can only observe the state of a patient if and only if they are treated? Cannot we monitor their state even if they are not under treatment? =========After reading reviews and authors' response========== I'd like to thank the authors fr their response. They fairly addressed my concerns. Hence, I'd like to keep ma score.


Review 2

Summary and Contributions: This paper introduces a new problem called Collapsing Bandits (CoB), which is a subclass of restless bandits (RMABs). Although the relaxed RMAB can be solved using a greedy index heuristic known as Whittle index, computing this index is typically slow and obtaining performance guarantees on such approach requires the problem to satisfy specific conditions (i.e. indexability). This paper shows that the CoB setting is both more general than previous models, and has the properties allowing for an efficient computation of the Whittle index and performance analyses. This is notably achieved using assumptions and structure on the transition probabilities between the states of each arm. Building on this, the authors propose an algorithm for computing the Whittle index. They characterize the performance of this algorithm using experiments on synthetic and real-world data.

Strengths: The assumptions leading to this subclass of RMAB seem reasonable and allow a large improvement in the computation of the Whittle index. The investigated setting is motivated by real-world applications. This is supported by an experiment on real-world data. This experiments is conducted against difficult baselines (oracle) and still highlight the performance of the proposed approach. Experiments on synthetic settings not only highlight the benefits of the proposed approach, but also its limitations, e.g. in situations where assumptions do not hold.

Weaknesses: The reduction of RMAB to arms with 2 states may be limiting in some cases. It is unclear how this could be relaxed to cases with M (finite) states per arms. The conclusion lacks openings and future works, which would help to motivate further research on the topic and acknowledge the limitations of the proposed setting and techniques.

Correctness: The claims and methods seem correct. The methodology in empirical experiments is correct.

Clarity: The paper is well written.

Relation to Prior Work: The RMAB setting introduced in this work is a more general subclass of RMAB compared with the other existing RMABs. Sec. 2 reviews previous settings and clearly explains how the current one differs. The papers also explains how the theoretical results (e.g. Thm. 1) generalize existing results. It seems like a link with combinatorial bandits (Chen et al., 2013) could be made considering that, on each episode, the agent selects of subset of k arms (among N).

Reproducibility: Yes

Additional Feedback: === Update == After reading the rebuttal and the other reviews, I have decided to keep my rating.


Review 3

Summary and Contributions: This paper studies a variant of the restless bandit problem which is motivated by the task of a nurse trying to decide which patients to check on in a clinical setting where they can only check on a subset of patients at once. This problem is formulated using an MDP to model the progression of each arm. It is assumed that when an arm is selected, the state is exactly observed, but uncertainty propagates through the unobserved arms. For this problem, the authors present conditions under which the policy is ‘indexable’, and show some further assumptions under which a computationally efficient algorithm can be derived. They illustrate good experimental performance of their approach.

Strengths: - The problem studied in this paper is well motivated from a practical perspective, and I have not seen this specific variant of the bandit problem studied before. - It is nice to see settings in which the problem is indexable, and also that in some of these settings, the indexes can be calculated quickly. - The experimental results in this paper are quite strong. They show that, in the examples considered in this paper, their algorithm runs significantly faster than previous approaches, and suffers from little drop in performance.

Weaknesses: - In this paper, the authors show indexability under certain conditions but do not explain why indexability is useful aside from saying that ‘performance guarantees’ hold only under indexability in line 156. The particular performance guarantees they are interested in are never mentioned nor shown. Perhaps some results explaining what guarantees you can get using indexability would be nice. - I am a bit confused by the conditions required for Theorem 1 to hold. To me it seems quite impractical to require that the optimal policy has a certain form. The conditions in Theorem 2 and 3 are nicer. It would also be helpful to give an example where these conditions hold. Also, what can be said when neither the conditions of Theorem 2 or 3 hold? - I find the assumption that P is known very restrictive. I’m not sure how realistic this assumption is in the motivating healthcare problem, nor is it clear to me how uncertainty in P can be immediately incorporated into their algorithms. There is some attempt to deal with this in Appendix F, but it was very difficult to understand what was actually done. In particular, what are the ‘parameters of each row’ over which a beta posterior is maintained? The key in the experimental results is also not clear. It would also be nice to see some theoretical results on the performance of the algorithms when P is unknown, perhaps incorporating some of the known performance guarantees on Thompson sampling seen in bandits or reinforcement learning. - This paper takes a more classical perspective than is commonly seen at NeurIPS, so perhaps there is an issue with it not being relevant to the community. However, I think it is beneficial for us to see these different perspectives and for the different communities to interact (so this is neither a strength nor a weakness of the paper, I just didn't know which box to put it in).

Correctness: - It would be good to see some confidence bounds on the experimental results. - In the experiments there is often some pre-processing done to the data which seems very ad-hoc and is not justified particularly well. - In Figure 5, what happens when 0% of the processes are forward threshold optimal (and also not backwards threshold optimal)? Does their algorithm still work? - Also in the experiments, it is not particular clear what the ‘intervention benefit’ is and how it is defined (a formula would be helpful). It would also be nice to see comparison of the total reward of each algorithm.

Clarity: The written English in the paper is, in general, very good. There are a couple of areas where readability could be improved: - there is a lot of switching between average reward and discounted MDP, at times this can cause confusion. - some notation is not clearly defined (e.g. what is Omega?).

Relation to Prior Work: I am not particularly familiar with prior work on computing the Whittle index, however, the authors do seem to be quite thorough and make a conscious effort to explain the similarities and differences between their work and prior work. Perhaps it might also be worth discussing some of the work on combinatorial bandits and whether these ideas can be extended to this setting.

Reproducibility: Yes

Additional Feedback: Some questions: - Theoretically, how does the computational complexity compare to prior work on computing the whittle index? What is the computational complexity of Qian et al [26]? - Experimentally, it can be seen that the proposed method is consistently marginally worse than Qian et al [26]. This is surprising since, as far as I understand, the theoretical performance is the same, if not better, than prior work. Can this be explained? ========= After rebuttal: Thank you to the authors for responding to some of my questions, and in particular for highlighting that several previous works on similar problems also assume P is known. In light of this I have raised my score. I hope that the promised clarifications and additional details appear in the final version of the paper.


Review 4

Summary and Contributions: Update after author response: I would like to thank the authors for the detailed response and for addressing my concerns. I am keeping my score unchanged. -------- In this paper, the authors propose a novel bandit setting, Collapsing Bandits, where the uncertainty about the state of an arm is fully resolved when observed. The setting is motivated by medical interventions, where whether a patient adhered to a regimen is fully-know when a nurse meets the patient. However, since a nurse can only see a few patients any given day, the decision-making problem becomes amenable to this new restless multi-armed bandit setting. The authors make theoretical contributions about the existence of an index-policy, as well as an algorithm for fast computation of the Whittle index.

Strengths: The paper is well-organized and the development of ideas is easy to follow. I had difficulty figuring out all the details of the various theorems (I didn’t go through the full supplementary document), but the overall ideas make sense to me. Maybe adding some further details to assist the reader with the proofs would be helpful, but I acknowledge that it can be difficult given the space constraints. Overall, the paper develops a general family of bandits and provides promising theoretical results.

Weaknesses: 1. The main performance metric “intervention benefit” needs to be more clearly defined — I am not sure what it exactly means, only that the higher the better. 2. Why are there no error bars if the trials were averaged over 50 runs? Especially since some of the averages are very close to each other, error bars will impart some idea of statistical significance of the results. 3. Minor formatting/grammatical errors: “fraction the runtime”, “a novel such class”, “must thus”, “display perform similarly”.

Correctness: Yes. To the best of my knowledge.

Clarity: Yes. Including some more details on development of the proofs will improve it further.

Relation to Prior Work: Yes.

Reproducibility: Yes

Additional Feedback:

[Author Response · NeurIPS 2020]

We sincerely thank the reviewers for their thoughtful and constructive feedback. We address specific questions below.

**[R1] (1) Assumption that workers learn states only when acting.** This is the scenario today for health workers
in Mumbai, India managing tuberculosis (TB) patients (this paper's direct motivation). A single worker monitors
adherence and delivers basic care to large cohorts of geographically distributed patients via person-to-person phone
calls; offline monitoring is unavailable. **(2) Determining arms to pull:** The Whittle index policy, defined by Whittle
[35], pulls the $k$ arms with the largest Whittle indices. We will make this explicit. **(3) Forward vs. reverse threshold**
**policies in simulations:** The majority of patients have forward threshold optimality, which we rely on in the simulations.
We will add details to the appendix. **(4) Comparison with Qian et al.:** For the optimality guarantees of the Whittle
index to hold for our algorithm, the process must satisfy the conditions of Thms. 1 and 2. However, the real world data
has a small fraction of patients who violate the condition of Thm. 2, resulting in the small gap in performance.

**[R2] (1) Extending from 2 to $M$ states:** The 2-state model is well-established in literature (Gilbert-Elliot model,
1960) and is popularly studied (e.g. seminal work of Liu and Zhao [19] that we extend) because of its wide range of
applications such as, to healthcare, anti-poaching, sensor maintenance, etc. Despite the wide applicability of this model,
generalizing to an $M$-state model will make for interesting future work. **(2) Future work:** We will add avenues of
future work to the camera-ready version. **(3) Link to combinatorial bandits:** Since RMABs also admit $\binom{N}{k}$ feasible
actions per round, this connection seems natural. However, in an RMAB, rewards on each sub-arm are state-dependent.
This would render existing combinatorial bandit algorithms – which maximize mean reward – sub-optimal in general.

**[R3] (1) Complexity:** Our work improves on the computational complexity of Qian et al., which has complexity per
round of $\mathcal{O}(Nlog(\frac{1}{\epsilon})(|\mathcal{S}|T)^{2+\frac{1}{18}})$. Our algorithm has a one-time cost of $\mathcal{O}(|\mathcal{S}|^2T)$ to precompute the Whittle indices
for all rounds, then has a per round cost of only $\mathcal{O}(Nmin\{k, log(N)\})$ to retrieve the top $k$ indices. We will make this
more explicit. **(2) Comparison with Qian et al.** Please see R1.(4). **(3) Indexability:** The guarantee that holds under
indexability is the asymptotic optimality of the Whittle index policy as proven by Weber and Weiss (1990) [33] referenced
on lines 39–40 of our paper. We will make this more explicit. **(4) Theorem conditions:** Thms. 2 and 3 give conditions
under which the structure required for Thm. 1 is theoretically guaranteed. Following are two examples of processes for
which conditions of Thm. 2 and Thm. 3 hold respectively (fwd: $P_{11}^a = 0.95, P_{01}^a = 0.9, P_{11}^p = 0.9, P_{01}^p = 0.4, \beta = 0.9$;
rev: $P_{11}^a = 0.95, P_{01}^a = 0.4, P_{11}^p = 0.4, P_{01}^p = 0.35, \beta = 0.9$). Since these are sufficient but not necessary conditions,
nothing can be concluded when neither is satisfied. However, we find from brute force checks that most processes,
even those that violate condition of Thm.2. are either forward or reverse threshold optimal. **(5) Assuming $P$ is known:**
This is realistic in many settings, as $P$ can be estimated from historical data collected either before or in early stages of
planning. E.g., in the TB domain mentioned in R1.(1), this data is gathered from health workers' early round robin
calling of patients. Further, since the offline planning portion of restless bandits is already PSPACE hard in general, it
is often studied separately from the online version (Liu and Zhao [19]; Meshram et al. [21]). Additionally, since the
optimal policy cannot be computed in general, regret bounds for general online restless bandit algorithms are typically
defined with respect to an arbitrary reference policy with full information, rather than with respect to the optimal policy
(e.g., Jung and Tewari [13]). This provides at least three reasons why developing strong algorithms for the version of
the problem with known P is of significant interest. **(6) Empirical methodology:** We have updated our figures with
confidence bounds (see Fig. 1 below). We have updated Fig. 5(d) of the main text to include $0\%$ threshold optimal
patients (Fig. 1(c) below); our algorithm shows strong performance. **(7) Preprocessing:** For the experiments derived
from real-world data, preprocessing only involved imputing missing action information to align with natural constraint
structure common in analogous domains (see lines 111–116). Further, sensitivity analysis in Appendix G confirms our
conclusions for a wide range of imputations. We will clarify this in the final paper. **(8) Intervention benefit** Please see
R4.(1). **(9) Link to combinatorial bandits** Please see R2.(3).

**[R4] (1) Intervention benefit** (described in text on Line 259) is calculated as: $I.B.(ALG) = \frac{\overline{R}^{ALG} - \overline{R}^{\text{No intervention}}}{\overline{R}^{\text{Oracle}} - \overline{R}^{\text{No intervention}}}$ where
$\overline{R}$ is the average reward of the algorithm as defined on Line 70. **(2) Error bars** Updated Figs. with error bars are below.

Figure 1: Error bars show difference in performance between our algorithm and Qian et al. is not statistically significant.

[Meta-Review · NeurIPS 2020]

The reviewers are all enthusiastic about the paper, though by varying degrees. The paper's main significant contribution is to the problem of planning for a class of partially observed restless bandits with two arm states each, for which a monotone transition probability structure holds -- the paper argues that this structure is quite natural in several applications, and demonstrates numerical results on one such setting involving medical interventions. It is shown that under this structure, the restless bandit is Whittle-indexable. Although there is no learning component addressed in the paper, the hope is that such a structural characterization will open up avenues for more work on learning good policies when there is a priori uncertainty about the restless Markov decision processes.